# Label-Free Fluorescence Molecular Beacon Probes Based on G-Triplex DNA and Thioflavin T for Protein Detection

**DOI:** 10.3390/molecules26102962

**Published:** 2021-05-17

**Authors:** Jun Xue, Jintao Yi, Hui Zhou

**Affiliations:** College of Chemistry and Chemical Engineering, Gannan Normal University, Ganzhou 341000, China; xuejun8657@foxmail.com

**Keywords:** label-free fluorescence, molecular beacon probes, G-triplex, thioflavin T, protein detection

## Abstract

Protein detection plays an important role in biological and biomedical sciences. The immunoassay based on fluorescence labeling has good specificity but a high labeling cost. Herein, on the basis of G-triplex molecular beacon (G3MB) and thioflavin T (ThT), we developed a simple and label-free biosensor for protein detection. The biotin and streptavidin were used as model enzymes. In the presence of target streptavidin (SA), the streptavidin hybridized with G3MB-b (biotin-linked-G-triplex molecular beacon) perfectly and formed larger steric hindrance, which hindered the hydrolysis of probes by exonuclease III (Exo III). In the absence of target streptavidin, the exonuclease III successively cleaved the stem of G3MB-b and released the G-rich sequences which self-assembled into a G-triplex and subsequently activated the fluorescence signal of thioflavin T. Compared with the traditional G-quadruplex molecular beacon (G4MB), the G3MB only needed a lower dosage of exonuclease III and a shorter reaction time to reach the optimal detection performance, because the concise sequence of G-triplex was good for the molecular beacon design. Moreover, fluorescence experiment results exhibited that the G3MB-b had good sensitivity and specificity for streptavidin detection. The developed label-free biosensor provides a valuable and general platform for protein detection.

## 1. Introduction 

Protein, an important component in human cells and tissues, participates in the physiological activities of the body, catalyzes the cell processes and plays important roles in the research of bioengineering, medical diagnosis, treatment and proteomics [1,2,3]. Some protein receptors can specifically bind to small organic molecules [4]. Such small molecular-protein receptor pairs have been widely used in drug delivery, molecular diagnostics and cancer therapies [5,6,7]. Therefore, the discovery and detection of small molecular-protein receptor pairs is significant and necessary. In most reported analytical methods, the terminal protection of small-molecule-linked DNA assays has attracted wide attention [8]. When the small molecule was bound to its protein receptor, the small-molecule-linked DNA was protected from the degradation by exonuclease. Additionally, the oligonucleotides of small-molecule-linked DNA not only act as coding sequences for identifying the linked organic molecules, but offer immediate signal amplification via polymerase chain reactions. This technology has encouraged many researchers to develop different analytical strategies to apply to the detection of H5N1 antibodies [9], FITC antibodies [10] and folate receptors [11,12]. However, these detection methods usually require expensive fluorescent labelling, complicated experiment treatment and precise detection instrumentation. Therefore, the development of simple and inexpensive methods avoiding dual-labelling fluorescence groups remains important for monitoring small molecule-protein interactions.

To address the challenge, some G-rich sequences are exploited in constructing label-free fluorescence biosensors, due to the G-rich sequence being able to self-assemble into a G-quadruplex structure. As we known, the G-quadruplex is a non-canonical DNA structure and noncovalently combines with small molecule ligands (such as thioflavin T or hemin) to form label-free probes, which have been widely used in biological progress monitoring and biomolecules analysis [13,14,15], wherein their applications occur with the concomitant conformational change of the G-quadruplex. However, the control and initiation of G-quadruplex structures are often faced with some difficulties. For example, in the design of the label-free molecular beacon (MB) based on a G-quadruplex, a long stem sequence usually contains many C bases and some complementary G bases, leading them to be hardly opened by the target DNA or RNA. Therefore, the design restraint is still a huge challenge for the application of label-free fluorescent probes based on a G-quadruplex in biosensors. 

With the deep study in the folding process of G-quadruplex structures, people find the G-triplex structure to be one of the most plausible intermediates formed by G-rich sequences with only three G-tracts [16,17]. What is more, the G-triplex structure displays catalyst functions similar to the G-quadruplex structure. With the advantage of concise sequences, the G-triplex structure has been widely applied in the design of molecular beacons (MBs). The MBs often suffer from some intrinsic limitations of the tedious and unavoidable sequence optimization, especially for those decorated with composite fluorescence or quenching groups, which greatly increases the experimental cost and complexity [18]. In order to solve these problems, our group previously developed a simple, universal and low-cost strategy for designing MBs [19]. The strategy was based on the stable G-triplex, which can combine with thioflavin T and act as a “signal-on” label-free fluorescence probe. For protein detection, the immunoassay based on fluorescence labelling has a good specificity but a high labelling cost. In this study, we developed a simple and label-free biosensor for protein detection based on the thioflavin T and G-triplex molecular beacon (G3MB). Biotin and streptavidin were used as model enzymes. In the presence of target streptavidin (SA), the streptavidin hybridized with biotin at the 3′-end of G3MB-b perfectly and formed larger steric hindrance, which hindered the hydrolysis of probes by exonuclease III (Exo III). In the absence of target SA, the Exo III successively cleaved the stem of G3MB-b and released the G-rich sequences, which self-assembled into a G-triplex and subsequently activated the fluorescence signal of thioflavin T. Compared with traditional G-quadruplex molecular beacons (G4MB), the G3MB only needed a lower dosage of exonuclease III and a shorter reaction time to reach the optimal detection performance.

## 2. Experimental Section

**Materials and Reagents.** The streptavidin, thioflavin T and exonuclease III were purchased from Sangon Biological Engineering Technology and Services Company Ltd. (Shanghai, China). Other reagents of an analytical grade were obtained from Sinopharm Chemical Reagent Company Ltd. (Shanghai, China) HeLa cells and MCF-7 cells were purchased from the cell bank of Xiangya Hospital (Changsha, China). Dulbecco’s modified Eagle’s medium (DMEM), RPMI 1640 medium, fetal bovine serum (FBS), streptomycin and penicillin were obtained from Thermo Fisher Scientific Inc. (Waltham, MA, USA) The ultrapure water (resistance > 18.25 MΩ) was obtained through a Millipore Milli-Q water purification system (Billerica, MA, USA). The DNA oligonucleotides were synthesized and purified by Takara Biotechnology Company Ltd. (Dalian, China), whose sequences were shown in Appendix A.

**Apparatus.** The fluorescence spectra were measured on the Fluorescence Spectrometer FS5 (Edinburgh Instruments, Scotland, UK). The excitation and emission slit were set at 5.0 nm, and with a 900 V PMT voltage. At room temperature, the CD spectra was carried on a Chirascan CD spectrometer (Applied Photophysics Ltd., England, U.K.). The gel electrophoresis images were captured by Tanon 2500R Imaging Analysis System (Shanghai Tianeng Company, Shanghai, China).

Fluorescence Measurements. In a typical assay to detect target SA, a mixture containing 100 nM G3MB1-b and different concentrations of target SA was incubated at 37 °C for 30 min. Then 0.5 U Exo III was added to the mixture and incubated for another 30 min at 37 °C. Subsequently, 6 μM ThT was injected into the reaction solution and incubated for 2 h at 37 °C before fluorescence measurement. All emission spectra were collected in the range from 460 nm to 600 nm at the excitation wavelength of 435 nm. All experiments were carried out at least triplicates. In the detection of SA, the concentration of G4MB3-b was 100 nM in one sample. 

**Gel Electrophoresis.** In the gel electrophoresis assay, a mixture containing G3MB1-b (1 μM) and SA (2 μg) in 1 X Tris-HCl buffer (25 mM Tris-HCl, 50 mM KCl, pH = 7.4) was first incubated at 37 °C for 30 min. Subsequently, 5 U Exo III was added to the mixture and incubated for another 30 min at 37 °C. Then 0.06 mM ThT was injected into the reaction solution and incubated for 2 h at 37 °C. The sample with 1 X loading buffer was applied to a non-denaturing PAGE (20%). The electrophoresis was carried out in 1 X Tris-borate-EDTA (TBE) buffer (90 mM Tris-HCl, 90 mM boric acid, and 2 mM EDTA, pH 8.0) at 150 V power for about 3 h at room temperature. After Stains-all staining and water eluting, the resulting gel was imaged with a Canon digital camera.

**CD Measurements.** The sample of G3MB1 or G31 (10 μM) was prepared in 1 X Tris-HCl buffer (25 mM Tris-HCl, 50 mM KCl, pH = 7.4) before CD measurements. Three scans were accumulated and collected from 200 to 500 nm, speed of 200 nm/min, bandwidth of 1.0 nm, and response time of 0.5 s. The optical chamber (1 mm path length, 150 μL volume) was deoxygenated with dry purified nitrogen before use and kept the nitrogen atmosphere during the detections. 

**Cell Culture Conditions.** HeLa cells were cultured in RPMI-1640 (GIBCO) supplemented with 10% FBS, streptomycin (100 μg mL^−1^), and penicillin (100 units mL^−1^). MCF-7 cells were cultured in Dulbecco’s modified Eagle’s medium with 10% FBS, penicillin (100 units mL^−1^), and streptomycin (100 μg mL^−1^) in a humidified atmosphere containing 5 wt %/vol CO2. 

**Application in Cells.** For the fluorescence measurements of G3MB sensors in cells, a mixture containing G3MB1-F (100 nM) with different concentrations of HeLa cells or MCF-7 cells was incubated in DMEM containing 10% fetal bovine serum for 2 h at 37 °C. Then, 0.5 U Exo III was added to the mixture and incubated for another 30 min at 37 °C. Subsequently, 6 μM ThT was injected into the reaction solution and incubated for 2 h at 37 °C.

## 3. Results and Discussion

**Design and Detection Principle**. In this study, we explored the use of a molecular beacon (G3MB) based on G-triplex in conjunction with a terminal protection assay for the enzyme amplification detection of protein receptors. We used biotin-streptavidin interaction as a model enzyme to demonstrate the feasibility of our method. As illustrated in Figure 1, the G3MB1-b is a hairpin structure probe with a small molecule of biotin attached to the 3′-end and possesses G-rich stem sequences at 5′-end. In the absence of target protein (streptavidin), the stem of G3MB1-b will be hydrolyzed successively into mononucleotides from the 3′-end by exonuclease III (Exo III), an enzyme that has specific 3′-to 5′-exonuclease activity for double strand DNA (dsDNA) in solution [20]. Then the released G-rich stem self-assembles into a G-triplex and activates the fluorescence signal of thioflavin T (ThT), a new small molecule ligand which can specifically bind to the human telomeric G-quadruplex [21]. In the present of a target protein (streptavidin), the streptavidin hybridizes with the biotin at the 3′-end of G3MB1-b, and Exo III fails to catalyze the stepwise hydrolysis of the biotin-linked G3MB1-b, leaving the G3MB1-b intact and thereby inhibiting the signal generation. 

**Feasibility of the G3MB1-b biosensor.** The feasibility of the G3MB1-b probe (a G-triplex molecular beacon probe with the biotin at 3′-end) for streptavidin (SA) detection was investigated by using fluorescence experiments. Some control experiments were first carried out. The G3MB1 probes (a G-triplex molecular beacon probe) were initially found to have weak thioflavin T (ThT) fluorescence (Figure 1A a, brown curve), due to the G3MB1 probe, which still kept its hairpin structure and the ThT was in a free state. When the G3MB1 was treated with 0.5 U Exo III, a significant increase in fluorescence was observed (Figure 1A b, yellow curve), because the stem of G3MB1 was degraded by Exo III (an enzyme had specific 3′-to 5′-exonuclease activity for double strand DNA) and the G-rich stem was released for folding into a G-triplex, which activated the fluorescence signal of ThT. In the present of the target streptavidin (SA), the significant decrease in fluorescence was observed (Figure 1A d, blue curve), because the SA could combine with the biotin at 3′-end of the G3MB1-b, as a result of great steric hindrance effect, the Exo III failed to catalyze the hydrolysis of the G3MB1-b, so G3MB1-b still kept hairpin structure and could not activate the fluorescence signal of ThT. In the absence of the target streptavidin, the G3MB1-b probe (biotin-linked-G3MB1 probe) incubated with 0.5 U Exo III, the system still exhibited a strong fluorescence signal (Figure 1A c, orange curve), due to no steric hindrance informed from streptavidin-biotin to influence the degradation of the G3MB1-b probe by Exo III, and the G-rich stem was released for folding into a G-triplex, which activated the fluorescence signal of ThT. Without biotin, the system still exhibited a similar strong fluorescence signal (Figure 1A e, green curve), due to no steric hindrance informed from streptavidin-biotin to influence the degradation of the G3MB1 probe by Exo III. Taken together, these results indicated that the G3MB1-b probe based on G-triplex DNA and thioflavin T could be used for streptavidin detection by the steric hindrance effect from streptavidin-biotin to protect the G3MB1-b probe against Exo III and cause the fluorescence change. In addition, the detection principle was further confirmed by using an agarose gel electrophoresis experiment, corresponding to the fluorescence experiments, as shown in Figure 1A. After incubating with Exo III, a new band (lane b) emerged with a faster electrical migration than that of the G3MB1 (lane a), due to the hydrolyzation of the G3MB1 stem. Although labelling biotin, the band (lane c) emerged the same as the band (lane b), indicating that the biotin had no influence on the hydrolyzation of the G3MB1 stem. In the present of SA, a new band emerged (lane d) near the hole, indicating the large molecular weight due to biotin–streptavidin interaction. Without biotin, the band (lane e) emerged similar as the band (lane b), due to no biotin–streptavidin interaction. These results were consistent with those obtained by fluorescence spectrometry. Furthermore, the detection principle was verified by a Circular Dichroism (CD) spectra experiment. As shown in Figure 1B, the CD spectrum of G31 (a G-triplex strand) had a strong positive peak at around 265 nm and a negative peak at 240 nm (green curve). However, the CD spectrum of G3MB1 had no obvious peak at 265 nm and a negative peak at 240 nm (orange curve), indicating that the G3MB1 kept hairpin structure and could not fold into G-triplex. When the G3MB1 stem degraded by Exo III, the CD spectrum exhibited a strong positive peak at around 265 nm and a negative peak at 240 nm (yellow curve), indicating the formation of G-triplex structure. All the above results demonstrated that the feasibility of G3MB in conjunction with terminal protection assay for the enzyme amplification detection of protein receptor.

**Sensitivity of the G3MB biosensor.** To obtain the optimal detection performance, the design of G3MB was first investigated. As shown Appendix AA, in the presence of different G3MB, the fluorescence intensity of ThT was detected. The fluorescence results displayed that the highest fluorescence intensity of G3MB0, and others all exhibited low fluorescence intensity. Considering the concise design and low fluorescence background of G3MB, we selected the G3MB1 for the subsequent experiments. In addition, the concentrations of Exo III were also investigated. As shown in Appendix AA, the fluorescence intensity increased remarkably with the addition of Exo III, and obtained a maximal value at 0.5 U Exo III (red curve). Furthermore, the reaction time was also investigated. As shown in Appendix AB, the fluorescence intensity increased quickly with the reaction time, and obtained a maximal value at 2 h (red curve). Therefore, the Exo III concentration of 0.5 U and a reaction time of 2 h were selected in the following experiments.

Under the optimized experimental conditions, the performance of the label-free G3MB1-b biosensor for protein detection was interrogated. With the increase of the target SA concentration ranging from 0 to 200 ng, the fluorescence intensity of the ThT emission peaked at 490 nm decreased gradually (Figure 2A). The fluorescence intensity of ThT increased linearly with the SA concentration in the range 0.1–50 ng (Figure 2B), and the detection limit was estimated to be 0.033 ng (in terms of the rule of 3 times deviation over the blank response), which was comparable to those reported protein detection methods [22,23].

The superiority of the G3MB was also investigated by comparing it to the G4MB. As shown in Appendix AB, the design of G4MB was first investigated. When the stem length of the molecular beacon was equal, all of the G4MB had a higher fluorescence intensity compared to G3MB, indicating the easy regulation and activation of G3MB. The fluorescence results showed the lowest fluorescence intensity of G4MB3. Considering the low fluorescence background of G4MB, we selected G4MB3 for the subsequent experiments. Compared to the G3MB1, the fluorescence intensity of the G4MB3 was lower at the same concentration of Exo III, and finally reached a maximal value at 0.5 U Exo III (Appendix AA, black curve). Compared to the G3MB1, the fluorescence intensity of the G4MB3 increased at a slower rate with the reaction time and reached a lower platform at 2 h (Appendix AB, black curve). Therefore, the Exo III concentration of 0.5 U and a reaction time of 2 h were selected in the following experiments. 

Under the optimized experimental conditions, the performance of the label-free G4MB3-b biosensor for protein detection was investigated. With the increase of the target SA concentration ranging from 0 to 200 ng, the fluorescence intensity of the ThT emission peak at 490 nm decreased gradually (Figure 3 B). The fluorescence intensity of ThT increased linearly with the SA concentration in the range 1–75 ng (Figure 3 B), and the detection limit was estimated to be 0.33 ng (in terms of the rule of three times deviation over the blank response), which was higher than the detection limit of the G3MB1-b biosensor. Taken together, compared to the traditional probe based on the G-quadruplex, the probe based on the G-triplex was easy to control and was excited. 

**Applicability of the G3MB biosensor.** The G3MB1-F biosensor is a hairpin structure probe with a small-molecule (folate) attached to 3′-end and possesses G-rich stem sequences at 5′-end. The feasibility of the G3MB1-F biosensor was investigated in the cells. As shown in Figure 3, the background fluorescence intensity of G3MB1-F was very low, indicating the superiority of G3MB1-F (red column). The fluorescence intensity increased remarkably with the addition of Exo III, due to the hydrolyzation of the G3MB1-F stem and the release of G-triplex DNA (black column). With the increasing numbers of HeLa cells, the fluorescence intensity remarkably decreased, due to the high expression level of folate-receptors in HeLa cells [24], indicating that the G3MB1-F biosensor could bind to HeLa cells and avoid the hydrolyzation by Exo III. However, the fluorescence intensity had few changes with the increasing numbers of MCF-7 cells, due to the low expression level of folate-receptors in MCF-7 cells [25], indicating that the G3MB1-F biosensor hardly combined with MCF-7 cells and was easily hydrolyzed by Exo III. All the above results verified that the G3MB1-F biosensor could be used for distinguishing cancer cells with different expression levels of folate receptors.

## 4. Conclusions

In summary, we have developed a label-free fluorescence G3MB biosensor based on G-triplex for protein detection. Biotin and streptavidin were used as model enzymes. With the target streptavidin, the biotin at 3′-end of G3MB1-b hybridized with the target protein and hindered the hydrolysis of probes by exonuclease III. Without the target protein, the stem of G3MB1-b is successively cleaved by exonuclease III, releasing G-rich sequences self-assembled into the G-triplex and activating the fluorescence signal of ThT. Due to the concise sequence of G-triplex, the G3MB needs a lower dosage of exonuclease III and a shorter reaction time to reach an optimal detection performance compared to traditional G4MB. Moreover, based on the interaction of folate and folate-receptors, the label-free fluorescence G3MB-F biosensor could identify cancer cells with different expression levels of folate-receptors. As a result, the proposed method holds great potential in protein detection and protein-associated biological study.

## Data Availability

The data presented in this study are available on request from the corresponding author.

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
