# Peer review of "Label-Free Fluorescence Molecular Beacon Probes Based on G-Triplex DNA and Thioflavin T for Protein Detection"

_molecules, 2021, doi:10.3390/molecules26102962_

Round 1

Reviewer 1 Report

Study of a molecular beacon based on G-triplex is presented. 

This paper demonstrates that attaching G3MB to biotin and later to streptavidin protects G3MB from degradation and therefore reduces fluorescence signal from ThT.

The results are presented in somewhat chaotic way (e.g. lines 167-170 are dementing themselfes, first indicating no role of SA, later a major role of SA). The results do not provide much explanation for the observed fluorescence reduction.

However, the paper presence a clear evidence, that G3MB interacts with some proteins that affect its fluorescence and could be therefore used as a kind of molecular beacon.

Author Response

Thank the reviewer for the helpful suggestion. We are sorry perhaps our description is not clear enough so as to induce the reviewer’s misunderstanding. The feasibility of the G3MB1-b probe (a G-triplex molecular beacon probe with the biotin at 3’-end) for streptavidin (SA) detection was investigated by using fluorescence experiments. In the present of the target streptavidin (SA), the significant decrease in fluorescence was observed (Fig. 1Ad, blue curve), because the SA could combine with the biotin at 3’-end of the G3MB1-b, as a result of great steric hindrance effect, the Exo III failed to catalyze the hydrolysis of the G3MB1-b, so G3MB1-b still kept hairpin structure and could not activate the fluorescence signal of ThT. In the absence of target streptavidin (SA), the G3MB1-b probe incubated with 0.5 U Exo III, the system exhibited strong fluorescence signal (Fig. 1Ac, orange curve), due to no steric hindrance informed from streptavidin-biotin to influence the degradation of the G3MB1-b probe by Exo III, and the G-rich stem was released for folding into a G-triplex which activated the fluorescence signal of ThT. The results indicated that the G3MB1-b probe based on G-triplex DNA and thioflavin T could be used for streptavidin detection by the steric hindrance effect from streptavidin-biotin to protect the G3MB1-b probe against Exo III and cause the fluorescence change. We have described this point more clearly in the revised manuscript (page 4, line -148 to line -172, sentences highlighted in red).

Reviewer 2 Report

The authors wish to report the development of a label-free fluorescence G3MB biosensor based on G-triplex for protein detection. To demonstrate the applicability of the proposed strategy, initially, biotin-streptavidin interaction was used as a model enzyme. Furthermore, folate and folate-receptors interaction was used to develop another label-free fluorescence biosensor which could identify cancer cells with different expression levels of folate-receptors.

The detection strategy is simple yet innovative, and it could be useful for further applications in various fields. So, the manuscript could be accepted for publication in this journal. A minor revision of the manuscript is required.

1. Mention the data collection / experimental conditions (solvents, pH, incubation time, temperature, etc.) in the respective figure captions.

Author Response

Thank the reviewer for the helpful suggestion. The data collection / experimental conditions (solvents, pH, incubation time, temperature, etc.) had been added in the corresponding figure captions (page 5, Figure 1, line -196 to line -199, line -200 to line -201, Figure 2, line -224 to line -227, Figure 3, line -257 to line -258, sentences highlighted in red).
